# Pet Loss in an Urban Firestorm: Grief and Hope after Colorado’s Marshall Fire

**DOI:** 10.3390/ani13030416

**Published:** 2023-01-26

**Authors:** Leslie Irvine, Casara Andre

**Affiliations:** 1Department of Sociology, University of Colorado, Boulder, CO 80309, USA; 2Front Range Veterinary Medical Reserve Corps, Aurora, CO 80016, USA

**Keywords:** pets, wildfire, animal rescue, evacuation, veterinarians

## Abstract

**Simple Summary:**

The Marshall Fire, a grass-fire-turned-urban-firestorm, destroyed over 1000 homes in southeastern Boulder County, Colorado, within six hours on 30 December 2021. The fire occurred on a weekday, when many residents were at work, and during the holidays, when many were traveling. When the fire began spreading rapidly in populated areas, roadblocks and dense smoke prevented people from returning home to rescue their pets. The fire displaced 30,000 residents. Although a precise count of animal deaths is not possible, it is likely that over 1000 pets died. Through interviews with pet owners whose animals died, this research examined what prevented them from rescuing their pets and what might reduce future mass animal fatalities. This research also assessed the fire’s impact on veterinary clinics located within the burn zone. The study challenges claims that attribute the failure to evacuate pets to weak human–animal bonds and adds to the literature on rapid-onset disasters.

**Abstract:**

Although much of the literature on pets in disasters associates the failure to evacuate pets with a weak or absent human–animal bond, rapid-onset disasters challenge the foundations of that claim. Colorado’s Marshall Fire, which occurred on 30 December 2021, took the lives of more than 1000 pets. The fire began in open grassland and quickly became an “urban firestorm” when it spread into densely populated areas. Due to the timing of the fire’s onset, owners could not return home to rescue their pets. Although first responders, volunteers, and other evacuees rescued some animals, many died inside their homes. Analysis of qualitative interviews with a small sample of pet owners whose animals died in the fire reveal the factors that prevented owners from rescuing their pets. Through analysis of traditional and social media, and emergency notifications, this research presents a timeline of events on the day of the fire and examines pitfalls in evacuation notification. Participant observation and field conversations provide insight into the impact of the fire on veterinary clinics. The study concludes with suggestions intended to reduce future mass deaths of animals.

## 1. Introduction

On 30 December 2021, the most destructive fire in Colorado history swept through southeastern Boulder County. The National Weather Service had released high wind warnings early that morning. Although high wind in itself is not unusual for Colorado, the forecast predicted extreme gusts of up to 90 miles per hour. In addition, atmospheric conditions known as “mountain wave amplification,” in which wind speed increases rapidly as air flows over the Rocky Mountains and down into lower elevations, pushed the wind even beyond forecasted expectations. Winds exceeded 100 miles per hour—strong enough to push several tractor trailers off of local highways [1]. In Colorado’s dry climate, high wind always brings the risk that any wildfire (or brush fire) that ignites will quickly spread. By December 2021, fire conditions were ideal. Abundant spring rain had produced lush vegetation, and exceptionally dry, warm weather in summer and autumn had left the area’s open grasslands critically parched. Around 10:00 a.m., firefighters responded to the first of three reports of brush fires. As they extinguished the first fire, a second grew large enough to necessitate temporarily closing an interstate highway. Upon extinguishing the second fire, all firefighting resources were directed to a third fire, reported at 11:00 a.m. At 11:21 a.m., fire crews reported locating “a small fire” and told the Boulder County dispatcher, “We’re extinguishing it now” [2]. Three minutes later, the crew informed dispatch that the fire was beyond their control. The wind drove the fire from open grassland into densely populated areas in the wildland urban interface. As the fire overtook the dense neighborhoods in the Town of Superior, the primary fuel source switched from dried vegetation to structures, and the fire quickly spread from home to home. Then, the fire pushed into the City of Louisville and unincorporated Boulder County. According to a report from the Colorado Division of Fire Prevention and Control, “Once the wildland fire had entered the urban environment, it no longer needed wildland fuels to propagate and had changed its character entirely” [3]. In addition, at this point, the fire’s intensity “increased exponentially” [3]. Officially known as the Marshall Fire, one climate scientist dubbed it an “urban firestorm” [4]. Within six hours, the Marshall Fire destroyed 1084 homes and damaged 149 more, along with 30 commercial structures [5]. It burned over 6000 acres. It displaced over 30,000 residents and 2 people lost their lives. The following morning, the wind decreased but temperatures plummeted, and snow began falling. By New Year’s Day, seven inches of snow covered the devastation. One year later, the cause of the fire remains undetermined.

Any incident that affects large numbers of people will also affect animals. As pets depend completely on people for daily needs, transportation, and escape from hazardous situations, they are particularly vulnerable in disasters [6,7,8,9]. As the Marshall Fire started on a weekday, and occurred between Christmas and the New Year, many residents were either working or traveling for the holidays. Many who were at home at the time managed to evacuate with their pets. However, road closures, dense smoke, and fast-moving flames prevented those who were not at home from re-entering their neighborhoods to rescue their pets. As discussed below, although the number of companion animals who died in the Marshall Fire can never be precisely determined, conservative estimates place it over 1000.

A sizeable body of literature documents the risks pets face in disasters. Pet evacuation understandably constitutes a prominent focus [6,7,8,10]. In some of the earliest research, Heath and colleagues warned that “pet ownership can be a significant threat to public and animal safety during disasters” [11] (p. 664). If a pet-friendly hotel, emergency shelter, or other accommodation cannot be secured in advance, attempting to find one during a disaster could delay evacuation [12]. In the absence of pet-friendly accommodations, owners may choose to ignore evacuation orders. For example, a survey performed by the American Kennel Club less than a year after Hurricane Katrina found that 62% of respondents would defy evacuation orders if they could not locate a place that would accommodate their pets [13]. Some owners might choose to leave their pets behind or be required—or even forced—to do so [14,15]. Leaving animals behind can jeopardize animal health [16,17,18]. It can also affect human mental health and emotional well-being [9,19,20,21]. As most pet owners in the United States consider their pets family members, the loss of a pet can result in significant psychological distress and trauma [17,22,23,24]. The experience can also have significant, lasting impact on children [25].

The failure to evacuate pets can lead owners to attempt to rescue pets left behind. In such circumstances, owners will reenter evacuated areas before disaster personnel declare the sites safe [17,18,26,27]. Research comparing evacuations following a flood and a hazardous chemical spill attributed 80% of unauthorized reentries to pet rescue [16]. Pet owners put not only themselves at risk, but they also endanger emergency personnel who might need to subsequently rescue individuals who reentered the area. Owners’ reasons for leaving pets behind at the time of evacuation often include thinking they would not be away from their homes for long. For example, in the chemical spill, emergency managers anticipated that the response would take several hours. Instead, it took over two weeks, reflecting the unpredictability of disaster response. Other reasons include not having a disaster plan, which leaves owners not knowing where to take their pets and being unable to transport them.

Some research associates the failure to evacuate animals with a weak bond, measured by levels of attachment and commitment to a pet [21,28]. Studies assess attachment and commitment to animals by indicators of care, such as visits to veterinarians and owning leashes or carriers. These studies find that owners with stronger attachments to their household pets are also more likely to have disaster plans inclusive of pets [11,29,30,31]. Conversely, a weaker standard of care indicates a weaker bond with an animal. People who leave their animals behind are those who keep their dogs primarily outdoors or who have no carriers available to transport their cats.

Other research reveals a more complex set of factors at work, however, and challenges the notion of a weak bond with animals. For example, following Hurricane Katrina, residents with lower incomes were more likely to leave animals behind during evacuation [10]. Moreover, many pet owners were forced to evacuate without their pets [6,32]. New Orleans residents who brought their pets to the Superdome—a designated evacuation site—were forced to leave them behind when they subsequently evacuated that facility because animals were prohibited on public transportation. Media accounts depict National Guardsmen simply letting dogs and cats run free as their guardians watch helplessly. One of the most famous images from the disaster shows a small white dog named Snowball being torn from a boy’s arms by a police officer as the boy boarded a bus to leave the Superdome [33]. Video showed the boy so upset that he vomited. The officer separated the dog and boy to uphold the policy that prohibits animals on public transportation. Evacuees reported being told that their animals would be rescued later, and some thought they could soon return for their animals themselves. Of course, some residents never returned. An estimated 200,000 pets were left behind in the aftermath of Hurricane Katrina [8,32]. It is unlikely that this number can be attributed to a weak human–animal bond.

In the Fukushima nuclear disaster, which followed an earthquake and tsunami in March 2011, pet owners faced a similar situation. The Fukushima Prefecture government gave residents mixed messages about what to do with their pets [17]. Pre-disaster instructions published in 2007 told pet owners to bring their animals with them, but instructions given at the time of the evacuation “did not allow residents to evacuate with their companion animals” [17] (p. 112). Officials initially told evacuees they would be gone for only a few days [18]. Instead, evacuees frequently moved from shelter to shelter as the evacuation zone widened, and some who evacuated with pets had to leave them behind when instructed to relocate. The number of dogs and cats left behind in the 20 km radius of power plant No. 1 is estimated to be between 10,000 and 20,000. Analysis of pet-related expenditures, such as veterinary care, food, and grooming, indicated that Japanese pet owners have a strong attachment to their pets. Thus, owners’ decisions regarding their animals at the time of the disaster do not stem from a weak bond [17].

In the United States, media coverage of the animals left behind following Hurricane Katrina, especially the footage of Snowball, drove the introduction of the Pets Evacuation and Transportation Standards Act, or PETS Act, in Congress. Signed into law on 6 October 2006, the PETS Act specifies that state and local disaster response plans should “take into account the needs of individuals with household pets and service animals prior to, during, and following a major disaster or emergency.” The Act has had mixed results. In assessing the response to Hurricane Irene, Hunt and colleagues conclude that “the media coverage around Hurricane Katrina and the subsequent PETS legislation have had positive effects on the evacuation of animals and that general awareness about the importance of evacuating pets has increased significantly” [34] (p. 537). However, Glassey’s interviews with individuals leading the animal response following Hurricane Harvey reveal that “only a minority [of responders] had specific knowledge of the PETS Act” [35] (p. 3). In sum, although the PETS Act represents a cultural shift in attitudes about the need to save animals in disasters, “the implementation of animal emergency planning appears sub-optimal and the integration of animal welfare charities to respond effectively remains fragmented in many areas” [35] (p. 3).

Although planning for animal care in disasters remains essential, the evolving circumstances of disasters often defeat even the best intentions. A joint analysis by the United Nations Office for Disaster Risk Reduction and the Centre for Research on the Epidemiology of Disasters reports that “disaster risk is taking on new shapes and sizes with every passing year” [20] (p. 3). This study is situated among others that focus on recent extreme and rapid-onset disasters [7,17,18,19,36]. These works examine disasters that defy the parameters of planned evacuation. With the goal of advancing both understanding and solutions, this study examines the inability to evacuate pets rather than the intentional failure to do so following a rapid-onset disaster. Our research questions were: (1) What prevented pet owners from evacuating their pets to safety? (2) What, if anything, might have helped them save their pets?

## 2. Materials and Methods

This research had three goals: (1) to conduct a census of lost, found, missing, and deceased pets; (2) to determine how to reduce pet casualties in future events; and (3) to assess the impact of the disaster on veterinarians and veterinary clinics within the burn zone. In pursuing these goals, this research faced logistical and methodological challenges common to studying disasters [37]. The sudden and unanticipated onset of events can prevent the design of research in advance. Community disruption can hinder representative sampling, and time constraints can limit the generalizability of findings. Additionally, because disasters continually evolve from onset through aftermath and recovery, data are often highly perishable [38,39]. People who initially agree to be interviewed relocate or circumstances change as they attempt to adjust to new realities. Nevertheless, research on disasters employs methods routinely used in social scientific studies, albeit in different contexts and with different constraints [39,40]. Consequently, this study draws on data from several sources frequently used in qualitative analyses. Traditional editorial and social media coverage constitutes the first source of data. This includes the primary regional newspapers (which publish in print and online), web posts, and Twitter feeds from the regional television stations, the Office of Emergency Management, and the Boulder County Sheriff’s Office. Media coverage was analyzed from the onset of the event through 15 January 2022, when coverage diminished and then stopped. In addition to news coverage, the media data set includes evacuation orders for the City of Louisville, the Town of Superior, and Boulder County from the morning of 30 December 2021, through 1 January 2022, when evacuations had ceased, and 11 pages of transcripts from the Boulder County Sheriff’s Office Marshall Fire Briefings, which are publicly available on YouTube. The media coverage was used to construct a timeline of the fire.

The second source of data includes pet owners’ posts and responses in the “Boulder County Fire Lost and Found Pets” Facebook group. Soon after the fire started, community members created this group, and within days, the group garnered over 20,000 members. Posts were collected with permission of the group moderator through May 2022, when posting about lost and found animals stopped. The posts allowed for a tentative count of lost and found animals. Moreover, the group served as a platform for recruiting pet owners for interviews, which constitutes the third source of data. Semi-structured interviews were conducted with eight owners whose pets died (or were thought to have died) in the fire. Interviews ranged in length from 30 to 60 min and occurred within two weeks of the event. Owners of lost pets were openly recruited from the Facebook group through an announcement about the research, posted with permission of the group’s moderator. For interviews, owners willing to participate clicked a link that led them to a scheduling app and consent form. All interviews were conducted either by phone or by Zoom. Participants did not receive compensation. Interviews began with a request for interviewees to describe, in their own words, how events unfolded for them the day of the fire. This allowed them to choose starting points and directions for the narratives of their experience. Established qualitative data analysis techniques including coding and memoing were used to analyze interview transcripts. Of the eight owners who responded, all but one were female. All were white, consistent with the majority of the population within the affected communities.

Interviews with two key staff members of the local animal shelter assigned primary receiving responsibility for found or deceased animals were also conducted. These interviews focused on numbers of animals received, their status on intake (e.g., healthy/injured/dead on arrival), their subsequent veterinary treatment, and reclaim status (e.g., owner identified and notified; owner unknown). In addition, the second author was a participant-observer within the veterinary professional community through the area’s Veterinary Medical Reserve Corps (vMRC). This provided access to information about affected animal care, need for medical supplies, and search and rescue support. Consequently, Zoom discussion groups were offered to veterinary professionals from affected clinics within the burn zone. An informal communication outlet was established through the Slack app to facilitate information exchange within the veterinary professional community. The Zoom discussion groups ranged from 60–90 min and included veterinarians, veterinary professionals from volunteer response groups, and other members of the veterinary community, along with pet loss grief counselors. Between two and five veterinarians participated in these virtual sessions. Similar to the pet owners interviewed, veterinarians and shelter staff members were also white and female.

## 3. Results

### 3.1. Rapid Onset Combined with Inconsistent Emergency Notifications

As mentioned above, many area residents were at work or out of town when the fire started. In interviews, residents who were at work said they learned of the fire primarily through media coverage or word of mouth. Some claimed not to receive emergency notifications or evacuation orders. By the time the news reached them, roads were congested with evacuees. Some, including major highways, were closed to all but emergency vehicles, making travel slow and some routes inaccessible. For example, one interviewee said that the drive from his workplace in the City of Boulder to his neighborhood in Louisville, which normally took 15-to-20-min, took nearly two hours that day. Upon reaching his neighborhood hoping to rescue his cat, he found his street was blocked off and he could not enter to see if his home was still standing. “I thought I’d have time to get home and get the cat, but nobody had access,” he said. “Everything was shut down”.

The media coverage and emergency notifications allowed for construction of a comprehensive timeline of the event (see Appendix A). These records shed light on evacuation delays resulting from pitfalls in emergency notification. At 11:47 a.m., the Boulder County Sheriff’s Office issued the first mandatory evacuation orders through the landlines, cellphones, and email addresses of residents who had opted into the warning system. After the fire, many residents said they did not know that they needed to opt into the emergency notification system to receive alerts on their cell phones. Of the 215 notifications sent, only 54 contacts confirmed receipt of the message as instructed. To be marked as confirmed, recipients of the message needed to press 1 if they received a reverse 911 call or click on a link if they received a text or email. Countless people likely received these notifications but dismissed the instructions, so the exact number of messages received is impossible to know [41,42,43]. Only twenty minutes after the Sheriff’s Office sent the first notifications, fire engulfed the first home in Superior. By 1:00 p.m., the entire town of Superior had been ordered to evacuate. Evacuation orders were soon issued for most of the city of Louisville and nearby areas of unincorporated Boulder County. However, only six percent of notifications were confirmed as received.

Analysis determined how many of the total notifications sent out to residents were actually received. Table 1 lists the success rate of each notification sent out by the Boulder County Sheriff’s Department. Of the 24,289 emergency notifications sent out by the Boulder County Sheriff’s office during the Marshall fires, only 4637, or 19%, were confirmed as received.

### 3.2. Accounts of Evacuation and Loss

Many residents were traveling between Christmas and New Year’s Eve. One of these pet owners knew nothing of the fire until she received an “Are you okay?” text from a friend. She subsequently watched the fire on her home’s security cameras as she and her husband drove back from another state. The security camera stopped recording and the screen said “Loud audible noise” before going black. “We figured we had done everything right,” she said, by leaving their two cats in the care of a trusted neighbor. Tragically, the neighbor’s home in Superior was among the first to go up in flames. After the fire, trained bloodhounds searched the site but found no scent. The owners also sifted the site hoping to find remains, but found nothing. A Louisville pet owner who was also away from home lost three dogs and a cat when the pet sitter left them behind. “He just got in the car and left,” she said. “He could’ve left the door open. He could have picked up our [small dog].” Neighbors watched him leave but did not know the pets were in the house. She emphasized how traumatic the loss was for her 14-year-old daughter, who had begun therapy as a consequence. She visited the site multiple times to sift for remains but had found nothing. “Despite losing everything,” she said, “it’s the pets we grieve”.

Some residents who were home at the time evacuated with their animals. However, even those who were at home could not necessarily evacuate their animals in time. One pet owner had noticed the wind and seen smoke in the distance but had not received an emergency notification. She was putting out her recycling, struggling with the containers in the wind, when a neighbor drove by, honked the horn, and shouted, “Get out! Get out!” The woman ran back into the house to get her three cats, but she could not get them into the carriers. She left the house, thinking she would be back later that day. When she tried to return, the road was blocked off. Road closures also prevented many residents who were not at home but were still in the area from re-entering their neighborhoods to rescue their pets. One interviewee had gone to check on her horse, whom she boarded at a stable near the site of the already-contained Middle Fork Fire. She had driven the family’s single car, leaving her husband at home with their two children, two dogs, two guinea pigs, and a cat. Unaware of the fire that was spreading while she had been at the stable, she was puzzled about the amount of traffic she encountered on her way home. She received a frantic phone call from her husband, stranded at home without the car. She had the phone on speaker and heard the smoke alarm blaring and her children screaming in the background. Her husband managed to get the children and the dogs out of the house, and a police patrol car rescued them from their smoke-filled neighborhood. The daughter had initially picked up the two guinea pigs intending to carry them with her, but she had set them down to get dressed. They ran off and were not recovered. Their cat was in the house but nowhere to be found when they had to leave. The woman said she felt sure that, had she been home at the time, she could have located the cat and gotten him out.

One family of four was visiting the area from out of state and were staying in an Airbnb in Superior. Understandably, they had not signed up for emergency alerts and were not following local news during what was to be a vacation. They left their two Labrador retrievers in the property for what they thought would be just a few hours. By the time they learned what was happening, the place they were staying had burned and both dogs perished in the fire. Bloodhounds from the non-profit organization Justice Takes Flight searched the area within a few days and found the remains of the dogs, next to each other, by their food bowl, scorched by flames. A picture of the bowl was circulated widely on social media (Figure 1).

In interviews and posts on the “Boulder County Fire Lost and Found Pets” Facebook group, residents who lost pets in the fire held on to hope that their pets escaped the flames and smoke when windows broke in homes. Trained rescuers and volunteers set up traps, trail cams, and feeding stations once they were authorized to enter the area on December 31. This effort was especially concentrated on cats. Soon after the fire, the renowned feline search and rescue expert Shannon Jay arrived to assist with locating missing “fire cats.” Jay had found hundreds of fire cats after California’s 2017 Tubbs Fire and the 2018 Camp Fire. Working with drone pilots, Jay had also rescued cats lost in the deadly 2021 Kentucky tornado. Jay brought an aerial thermal imaging device and a ground-level handheld thermal imaging scope to the task in the Marshall Fire. Despite spotting a number of cats, especially at night, Jay stated, “As I gathered intel on the situation, we came to understand that the cast [sic] majority of the missing felines were known to be INSIDE their homes when they were burned to nothing. Based on my experience and educated guess, I opine that the odds of a firecat making it out of a burning home is about 5%” [44].

One cat who did make it out of a home in Superior became a social media star. Merlin, a 9-year-old tabby, was home on December 30. His owner had gone to his job in an area where he did not have cell phone service. That evening, when he had service again, he found a long list of messages, with one from a neighbor telling him, “It’s all gone” [45]. He had lost everything. The owner recalled, “The only thing that went through my mind was Merlin. I just felt broken. Like someone just ripped my soul out” [45]. The next night, rescuers found a badly burned cat meowing on the porch of the sole house left standing on the block [45,46] (see Figure 2). They rushed the cat to a local veterinary clinic for emergency care, including fluids and oxygen. The cat’s burned fur initially made it impossible to tell the color or markings and even the sex, but a microchip allowed them to locate the owner. When reunited with his owner at the veterinary clinic, rescuers reported that Merlin started purring as soon as he heard his owner’s voice [47]. Merlin was hospitalized for over a month. His story spread rapidly on social media as devoted followers tracked his progress. He has made a full recovery.

### 3.3. Census of Lost Pets

Two factors ruled out making an accurate count of animal fatalities. The first is the absence of a comprehensive pre-disaster census of owned animals. Although the City of Louisville requires residents to license dogs, neither the Town of Superior nor unincorporated Boulder County have a licensing requirement. At the time of the fire, 618 dogs were licensed in Louisville; however, not all dog owners license their dogs, even when required by law. None of the communities of Louisville, Superior, and Boulder County require licenses for cats. Thus, no official, baseline pet ownership data exists. In addition, the absence of a comprehensive reporting system for animals missing in the fire constitutes the second obstacle to a census. The “Boulder County Fire Lost and Found Pets” Facebook group was the main site for reporting lost pets after the fire. Using it requires a Facebook account and the time and ability to post a report. Despite these limitations, survey data from the American Veterinary Medical Association (AVMA) allows for an informed estimate of the number of lost pets. According to AVMA pet ownership surveys, 60% of Colorado households include dogs, cats, birds, and horses [48]. Moreover, most pet-owning households include more than one animal. Using the AVMA’s count, of the 1233 homes destroyed or damaged in the Marshall Fire, 739 of these households had at least one pet. Estimating the number of households with additional pets raises the potential number of animals affected to 1182.

Of course, not all of these animals died in the fire. Some animals were rescued by first responders, volunteers, or other evacuees who took them to local animal shelters, where those who suffered burns and other injuries received treatment. Table 2 reports the numbers of dogs received by the Humane Society of Boulder Valley (HSBV). In addition, 24 strays, mostly dogs, were brought in and all were reunited with their grateful families. Only five animals were brought in dead on arrival. HSBV encouraged pet owners to submit “lost” reports in an effort to match with animals brought in (see Table 3). They reported receiving 107 reports. Of these, 26 were cancelled through reuniting with owners and 24 were cancelled through confirming death. The remaining 57 lost reports remained open one month after the fire.

Among the eight residents interviewed for this research, 17 pets were lost and assumed deceased in the fire (10 cats; 5 dogs; 2 guinea pigs). Overall, it appears that the majority of animals at home at the time of the fire were not rescued. As mentioned, some owners found the remains of their animals in their devastated property, but the fire’s intensity combined with the heavy overnight snowfall immediately after the fire reduced the likelihood of recovering remains. One experienced responder described the fire as a “‘mass casualty event’ for pets who were in homes that burned” [46]. Another posted this on Facebook: “One cannot truly fathom how absolutely heartbreaking, devastating and overwhelming something like this is. Many of us have been in disaster response for several years and have done hurricanes, tornadoes, earthquakes, hoarding cases and seen pretty much everything bad there is to see in the animal world. This was absolutely the worst.” [47].

### 3.4. Impact on Veterinary Medical Professionals

The second author had access to consolidated information about the impact of Marshall Fire on the veterinary professional community through the area’s Veterinary Medical Reserve Corps (vMRC). As an organization with connections to the local veterinary community, the vMRC collaborated with area professional organizations, such as the Colorado Association of Certified Veterinary Technicians and the Colorado Veterinary Medical Association, and other volunteer and animal care groups to provide information about affected animal care, need for medical supplies, and search and rescue support. Unfortunately, communication between disaster response organizations working within the disaster zone and the veterinary medical community was fractured and slow. Veterinary medical professionals did not know how to access official information outlets. Consequently, various groups on social media platforms quickly arose to allow communication between area veterinary professionals. Lack of early, consistent communication between government and volunteer groups resulted in multiple self-deployments of veterinary professionals during the fire and in the initial weeks of recovery, when the search for lost animals was active. If communication had been better, and informed by pre-event planning, veterinary professionals could have reduced redundancy in the search and recovery efforts. Moreover, they could potentially have provided a unified source of records about the locations of missing pets.

Data collection on the needs of displaced area pet owners was attempted through clinic records but with minimal success. Anecdotal reports shared among area professionals suggest that many clinics provided food, medication replacement, and initial medical care at no cost to clients. Unfortunately, clinics did not track the provided services and supplies effectively enough to report.

The veterinary clinics in the affected area came through the disaster with few physical losses. During the initial weeks following the fire, neighboring clinics shared staff members and clinic functions so that all clinics could reopen. However, it is important to note that most veterinary clinics were operating at or beyond capacity prior to the Marshall Fire. A backlog of non-essential procedures from the COVID-19 pandemic combined with the unprecedented influx of new pets to households overwhelmed the resources of many clinics. The addition of urgent visits in the aftermath of the Marshall Fire stressed area clinics even further. Moreover, clinic staff members suffered trauma from damage to or loss of their homes, damage or loss to neighborhoods around the clinic, from the loss of their patients, and the grief experienced by their clients. The vMRC offered three debrief/after-action review sessions to the veterinary medical community. Virtual discussion sessions offered conversations between colleagues about the experience and memorializing the event. Over the course of the three-event series, 16 individuals from the vMRC team, area clinics and shelters, pet loss counselors attended the sessions. In the immediate weeks after Marshall Fire, Colorado State University’s Argus Center and local pet loss support resources provided mental health support to area clinics and shelters. Mental Health Partners, Care for the Healer, Mental Health Partners, and the Colorado Spirit team provided longer-term, embedded mental health support during a 6-week pilot period exploring the best ways to provide post-disaster support to medical professionals serving affected communities. After the fire, the weather continued to be dry and carry a high fire risk. Consequently, area veterinary clinics reported a loss of staff members due to anxiety about the potential for another fire.

## 4. Discussion

This research has at least two limitations. First, although the analysis focuses solely on pets, “pets” can be defined in many ways. Strictly speaking, the term refers to those animals that are given individual, personal names, live in the house, and most importantly, are never eaten [49]. Any species of animal can qualify as a pet, the most common pets in western societies today are dogs and cats, followed by birds and specialty or exotic pets, such as fish, ferrets, hamsters, guinea pigs and other rodents, snakes and other reptiles, and amphibians. Horses can be considered pets, but their simultaneous status as “large animals” or livestock makes them better categorized as “border species” [50]. On farms, favored cows, sheep, or pigs can be named and treated as “petted livestock,” rather than sent to slaughter [51]. This analysis does not include all species that were affected by the Marshall Fire. It does not include the fire’s impact on horses and livestock. Although news media covered several horse rescues, no horse fatalities were reported. This analysis does not include the fish and other animals who died in aquariums when homes lost power, nor—apart from the two guinea pigs mentioned above—does it include the many small animals who perished. Thus, the estimates of animal deaths reported here would be higher if records of other species were available.

The small size of the sample of interviewees constitutes a second limitation. In most research, only tentative conclusions can be drawn from a sample size of eight people. However, qualitative studies usually employ purposive samples (rather than probability samples), which provide information-rich sources. The question of sample size for qualitative researchers is less a matter of the number of participants and more one of “saturation.” This refers to a point at which data collection reveals no new insights and information becomes redundant. In this study’s interviews, all eight pet owners discussed the inability to return home, either because of absence from the area or emergency closures. All eight discussed pitfalls with emergency notifications. Although additional interviews might have provided further details about individual circumstances, the goal of learning what prevented people from rescuing their pets was attained through a limited number of interviews.

This research serves as a reminder of the need to include pets in emergency planning for households. Information on such planning is widely available online. However, even the most comprehensive plans would have been of little use in the Marshall Fire because so many of the area’s residents were away from home. Future research needs to explore pitfalls in the emergency notification system, which prevented many pet owners from returning home in sufficient time to rescue their animals. In addition, research should examine the evolving role of social media in locating lost pets in disasters. As DeYoung and Farmer point out, one of the biggest changes since Hurricane Katrina is the presence of social media as “a space for animal welfare advocates, experts, community members, emergency managers, and other actors to communicate, share information, and coordinate during and after disasters” [7] (p. 92). Social media will continue to serve this role, and research can assess its effectiveness, inform its development, and ensure its legitimacy.

One benefit of increasing the likelihood that pets will be rescued is the decrease in the experience of grief associated with pet loss. All interviewees expressed feelings of sadness, depression, helplessness, anger, and guilt related to the inability to save their pets. One interviewee told us her teenage daughter had already begun therapy to cope with the trauma. Moreover, their grief was often “disenfranchised,” or underestimated and unrecognized in the context of the overall tragedy and all that recovering from a disaster requires [52].

The emotional toll accompanying pet loss is the counterpart to the joy and other benefits that brought by their companionship. Even in situations of “normal” pet loss, such as death due to age or illness, the bereavement can involve symptoms of depression three times greater than those typically found in the wider population. [53] Bereaved pet owners also report feelings of anger, guilt, loss of sleep, loss of appetite, and disruption of daily activities. [54,55] Pets are widely regarded as family members, and levels of grief experienced after pet loss can equal those that follow the loss of human family members. [56] Research on the impact of pet loss in disasters associates the experience with post-traumatic stress, depression, and other symptoms. [23,24,25] Indeed, the mental health impact of pet loss following a disaster has been found to exceed the emotions resulting from being displaced from one’s home. [23]. Following the Marshall Fire, pet owners coped with their grief privately in various ways. Importantly, public events, both immediately following the fire and planned for the future, openly acknowledge the tragic loss. In January, the Humane Society of Boulder Valley and a Boulder church held vigils for lost pets. [57] At the time of this writing, a Louisville non-profit organization is raising funds for a public art installation in memory of pets who died in the fire. [58] Efforts such as these recognize and validate the grief felt by pet owners and provide valuable social support by allowing them to mourn openly.

Several web-based apps for pet rescue in disasters are available or in development. They allow pet owners to post notifications about pets left in homes and in need of rescue. However, this technology ultimately relies on human relationships. The effectiveness of a rescue app requires people who can enter one’s home in an emergency. Moreover, pets should also trust these people enough to allow them to enter their home, handle them, leash them, put them in carriers, and transport them as needed. For instance, one Superior resident’s dog was home, in a crate, when the Marshall Fire approached the neighborhood. A neighbor knew the resident was not home. As the neighbor was evacuating, he went to the house, kicked in the door, and rescued the dog, who certainly would have died. News coverage shows the neighbor putting the dog in his car. [59] Thus, social connections may be the best way to save the life of a pet.

## 5. Conclusions

Although the literature has tended to associate the failure to evacuate pets with a weak human–animal bond, the circumstances surrounding the spread of The Marshall Fire suggest that other factors need to be considered. In particular, researchers may have over attributed the role of the human–animal bond in their analyses, especially when examining decisions to leave animals behind, and failed to account for the contextual dynamics of the event. The Marshall Fire’s intensity and behavior, coupled with pitfalls in the emergency notification system, rather than lack of care for or attachment to animals, resulted in high pet fatalities. The vulnerability of animals in disasters is especially evident among cats. Countless cats who could not be caught or located were left behind, and many lost their lives. In the Marshall Fire, as in Hurricane Katrina and the Fukushima nuclear disaster, factors well beyond the control of pet owners prevented many from saving their animals. In its rapid onset in a highly populated area, the Marshall Fire was an unprecedented event in Colorado. Densely populated neighborhoods are not typically vulnerable to wildfire, but the Marshall Fire showed how quickly a wildfire can become an urban firestorm. Contextual dynamics, such as the speed and intensity of the fire, combined with physical distance, led to the loss of animals’ lives.

## Figures and Tables

**Figure 1 animals-13-00416-f001:**
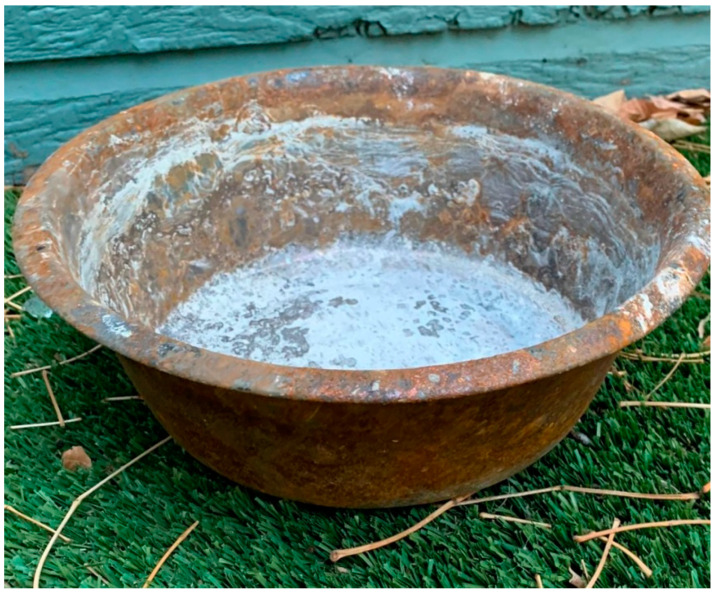
Stainless steel dog bowl scorched in the fire. (Photo credit: Patti Benninghoff-Lawson).

**Figure 2 animals-13-00416-f002:**
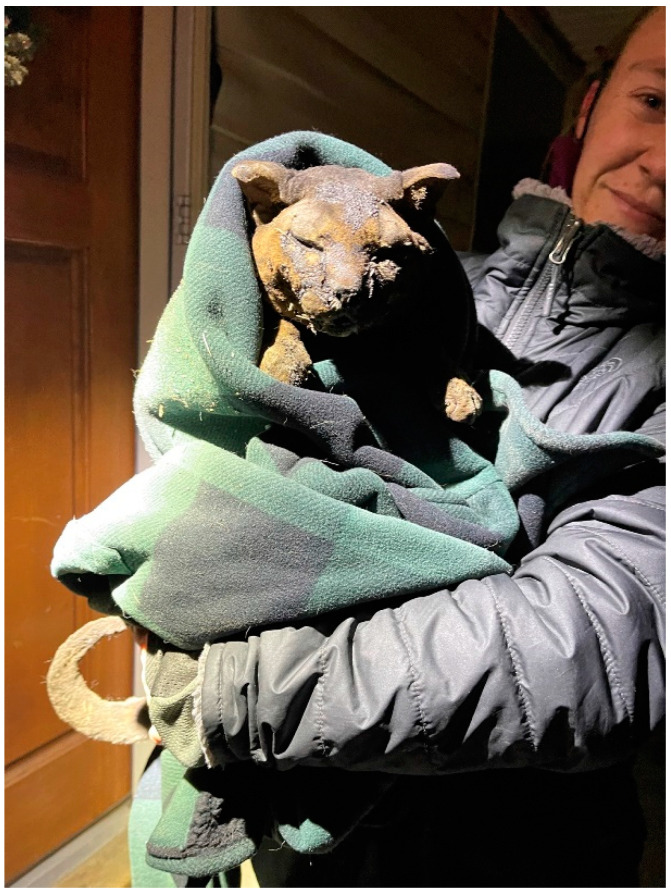
Merlin at the time of rescue. (Photo credit: Shelby Davis, Soul Dog Rescue).

**Table 1 animals-13-00416-t001:** Notifications from Boulder County’s Emergency Notification System.

Date 12/30/2022	Total Contacts	Received on Time	Received Late	Unreachable	Unconfirmed	Success Rate
11:47:52	215	54	2	4	155	25.1%
12:15:55	2588	150	7	6	2425	5.8%
12:46:18	254	39	5	1	209	15.4%
12:49:56	4173	1010	34	35	3094	24.2%
13:08:48	7251	706	14	61	6470	9.7%
13:15:08	2509	202	1	12	2294	8.1%
13:25:39	276	128	2	2	144	43.4%
14:51:14	4806	1519	55	86	3146	31.6%
14:58:04	2217	829	26	14	1348	37.4%
Total	24,289	4637	146	221	19,285	19.1%

**Table 2 animals-13-00416-t002:** Animals Received at the Humane Society of Boulder Valley.

Received from owner evacuees for boarding	33
Stray	24
Dead on arrival	5
Subtotal animals received	62
Strays reclaimed by owners	24
Total animals received	38

**Table 3 animals-13-00416-t003:** Lost Animal Reports Filed at the Humane Society of Boulder Valley.

Total reports filed	107
Reports cancelled through reuniting	26
Reports cancelled; confirmed deceased	24
Reports remaining active	57

## Data Availability

The data presented in this study are available on request from the corresponding author. The data are not publicly available due to privacy reasons.

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
