# Peer review of "Pet Loss in an Urban Firestorm: Grief and Hope after Colorado’s Marshall Fire"

_animals, 2023, doi:10.3390/ani13030416_

Round 1
Reviewer 1 Report
Article Review: animals-2145045
This study is an important contribution to the literature on pets and disasters. Not only does it effectively challenge the centrality of the human-animal bond in the literature, but it also extends the examination of pets and disasters into the context of rapid onset events. It is a well-developed multi-method approach to examining the entire event, and not just in isolation of pet ownership from the larger context.
My only recommendation would be to consider a rephrases the conclusion, especially the last sentence in the conclusion. I am not certain that there is enough evidence in this study to make the current final statement in the paper with such certainty. (One example of a potential lack of bond was the pet sitter leaving the pets in their charge to die.) Based on the authors’ arguments about forced separation from pets during the Hurricane Katrina evacuation, Fukushima nuclear disaster, etc., and the results presented in this paper, it might be more appropriate to consider that researchers have over attributed the role of the human-animal bond in evaluations, especially decisions to leave animals behind, and failed to account for the contextual dynamics of the event. It is one of the most important contributions of this study. The conclusion needs to be more nuanced and direct future research to better examine the complexity of pet ownership and disaster response, and how the event and response, despite bond, could put animal lives in danger.
Author Response
Thank you for making this suggestion. We agree that we do not have sufficient evidence about human-animal bonds to make the claim we made in the previous draft. In this revision, we have taken the liberty of using your phrasing to point out that the circumstances and dynamics of the fire are the primary reason for the failure to evacuate pets. The strength of the human-animal bond is secondary (and unexamined here).
Reviewer 2 Report
Very important study on a topic that is not often discussed-- rapid onset disasters and pets, as well as grief and loss of those pets. Some minor points:
What media were looked at for methods? Any specific method for including/excluding media sources?
page 9, line 396-- I think this could be fleshed out a bit more, if possible. This seems to be a point that comes up in research on pets and disasters quite often, that communication is lacking between government officials and volunteer groups. What might this have changed in this scenario, if communication was better?
In the discussion, I would like to see a bit more on the concept of grief and loss of pets. This is mentioned earlier in the paper, and of course the title eludes to this as well. While research on pets and disasters tends to focus on the immediate issues of evacuation, sheltering, and recovery-- I think one of the highlights of this research, and one important factor that other studies have not delved into, is on dealing with the loss of pets after disaster.
Author Response
Thank you for these insightful comments. As for the question about media, we have now specified that the sources include regional newspapers (print and online) and television stations (online), along with Tweets by the Office of Emergency Management and the Sheriff's Office. We have added a sentence clarifying that we used the media coverage to construct the timeline of the fire (which we again say in the Results, below).
On veterinarians and communication: we have added some ideas, such as reducing redundancy in the search-and-rescue efforts and potentially providing locations of missing animals.
You are absolutely right that we needed to devote more attention to grief. We have now done so in two new paragraphs in the discussion.
Reviewer 3 Report
Anyone who has a pet will have an emotionally hard time reading this detailed account of the events and deaths. It is a story that needs to be told. I think your conclusion are fully justified within this fact pattern. I am not sure there are any structural or administrative answers that could have changed the results when time is impossibly short to think and deal with the events.
Author Response
Thank you for your supportive feedback.